# Combination of Blood Adiponectin and Leptin Levels Is a Predictor of Biochemical Recurrence in Prostate Cancer Invading the Surrounding Adipose Tissue

**DOI:** 10.3390/ijms25168970

**Published:** 2024-08-17

**Authors:** Atsuto Suzuki, Shinya Sato, Noboru Nakaigawa, Takeshi Kishida, Yohei Miyagi

**Affiliations:** 1Morphological Analysis Laboratory, Kanagawa Cancer Center Research Institute, Yokohama 241-8515, Kanagawa, Japan; suzuki.8o60n@kanagawa-pho.jp; 2Department of Urology, Kanagawa Cancer Center, Yokohama 241-8515, Kanagawa, Japan; 3Department of Pathology, Kanagawa Cancer Center, Yokohama 241-8515, Kanagawa, Japan; 4Molecular Pathology and Genetics Division, Kanagawa Cancer Center Research Institute, Yokohama 241-8515, Kanagawa, Japan

**Keywords:** adipokine, adiponectin, leptin, prostate cancer, biochemical recurrence, castration-resistant prostate cancer, adipocytes, androgen receptor

## Abstract

Biochemical recurrence is a process that progresses to castration-resistant prostate cancer (CRPC) and prediction of biochemical recurrence is useful in determining early therapeutic intervention and disease treatment. Prostate cancer is surrounded by adipose tissue, which secretes adipokines, affecting cancer progression. This study aimed to investigate the correlation between blood adipokines and CRPC biochemical recurrence. We retrospectively analyzed the clinical data, including preoperative serum adipokine levels, of 99 patients with pT3a pN0 prostate cancer who underwent proctectomy between 2011 and 2019. The primary outcome was biochemical recurrence (prostate-specific antigen: PSA > 0.2). We identified 65 non-recurrences and 34 biochemical recurrences (one progressed to CRPC). The initial PSA level was significantly higher (*p* = 0.006), but serum adiponectin (*p* = 0.328) and leptin (*p* = 0.647) levels and their ratio (*p* = 0.323) were not significantly different in the biochemical recurrence group compared with the non-recurrence group. In contrast, significantly more biochemical recurrences were observed in the group with adiponectin < 6 μg/mL and Leptin < 4 ng/mL (*p* = 0.046), initial PSA > 15 ng/mL, clinical Gleason pattern ≥ 4, and positive resection margin. A significant difference was also observed in the multivariate analysis (hazard ratio: 4.04, 95% confidence interval: 1.21–13.5, *p* = 0.0232). Thus, low preoperative serum adiponectin and high leptin levels were significantly associated with biochemical recurrence in adipose tissue-invasive prostate cancer, suggesting that they may be useful predictors of biochemical recurrence. Further studies with larger cases are needed to increase the validity of this study.

## 1. Introduction

Prostate cancer has a particularly high incidence rate among cancers affecting men. In the United States, it is the leading cancer among men in terms of incidence and the second leading cause of death [1]. In Japan, its incidence has steadily increased with the aging of the population and Westernization of diets, and in 2019, it will be the leading cause of cancer in men.

Prostate cancer grows in an androgen receptor signaling-dependent manner and can be effectively treated by androgen deprivation therapy (ADT) [2,3]. However, when ADT is ineffective and castration-resistant prostate cancer (CRPC) develops, the primary tumors may spread and metastasize, leading to adverse outcomes [4,5]. The survival time of patients with CRPC is approximately 2–3 years. Although the number of available drugs has increased in recent years, the treatment effectiveness varies from patient to patient, and more effective treatments are needed [6,7,8]. The mechanisms underlying CRPC development remain unclear, although sustained activation of the androgen receptor (AR) signaling by its genetic changes and variants has been reported [9,10,11]. Therefore, there is an urgent need to elucidate the novel mechanisms of castration resistance acquisition that may serve as therapeutic targets.

We focused on studying the roles of adipose secretory factors in castration resistance acquisition. The prostate is surrounded by adipose tissue, an endocrine organ that secretes growth factors and adipokines [12,13,14]. Epidemiologically, obesity is also associated with prostate cancer progression, and metabolic syndrome is correlated with the prognosis of metastatic prostate cancer [15]. Our previous study showed that the microenvironment in which fat is present enhances the expression of castration-resistance-related genes in cancer specimens from patients [16]. It has been suggested that adipose-secreted factors are involved in cancer progression. Adiponectin and leptin are adipokines with relatively detailed mechanisms and abundant epidemiological data, especially in cancer cell growth [17,18,19]. In general, adiponectin inhibits cancer cell growth [20,21,22] and leptin promotes cancer cell growth [23,24]. In prostate cancer, adiponectin also inhibits cancer cell growth [25,26] and leptin inhibits cancer cell growth [27,28,29] as well. The combined clinical significance of adiponectin and leptin in prostate cancer patients has been studied in various contexts, including the risk of prostate carcinogenesis risk [30], patient obesity and abnormal glucose metabolism [31,32,33], adipokine expression in prostate tissue [34], and correlation with cardiovascular disease [35]. However, this is the first study to combine adiponectin and leptin as predictive markers of biochemical recurrence, aiming for a more accurate prediction of this outcome.

This study used clinical and pathological information and preoperative sera from patients with adipose tissue-invasive prostate cancer to examine how blood adipokine levels correlate with biochemical recurrence and progression to castration resistance in adipose tissue-invasive prostate cancer. We aimed to determine the relationship between adipokine and cancer recurrence in prostate cancer cases with the invasion of periprostatic adipose tissue, where cancer and fat interact and adipokines are directly exposed to cancer cells. The findings of this study could provide insight into developing new strategies for improving prostate cancer treatment.

## 2. Results

First, we examined the correlation between biochemical recurrence and blood adipokine levels in patients with pT3aN0M0 stage prostate cancer, excluding the influence of major prognostic recurrence factors, such as lymph node metastases. The patient background and selection flow are shown in Table 1 and Figure 1, respectively. There were 34 and 65 patients in the biochemical recurrence and non-recurrence groups, respectively.

No significant differences were observed in age, surgical technique, BMI, clinical T stage, clinical Gleason score, medical history, preoperative hormone therapy, or the number of deaths between the two groups (Table 2 and Table 3). The preoperative PSA level was significantly higher in the biochemical recurrence group than in the non-recurrence group (*p* = 0.006) (Figure 2), and the follow-up period was significantly longer in the biochemical recurrence group (*p* < 0.001) than in the non-recurrence group (Figure 3) (*p* < 0.001) (Table 2). No significant differences were observed between the two groups in the pathological Gleason score, resection margin (RM), lymphatic vessel invasion (ly), venous invasion (v), or perineural invasion (pn) (Table 3).

In the biochemical recurrence group, 26 patients (75%) underwent salvage radiation therapy, and one patient (2.9%) progressed to CRPC (Table 3). No significant differences were observed in serum lipid levels between the two groups regarding preoperative blood TG, T-chol, HDL, and LDL concentrations (Table 4). Furthermore, no significant differences were observed in preoperative blood adiponectin and leptin concentrations and adiponectin/leptin ratio (A/L ratio), but adiponectin (<6 μg/mL) and leptin (>4 ng/mL) were significantly higher in the biochemical recurrence group than in the non-recurrence group (*p* = 0.046), respectively (Table 4).

No significant correlation was observed between the time for biochemical recurrence and serum adiponectin and leptin levels, A/L ratio, and the presence of adiponectin (<6 μg/mL) and leptin (>4 ng/mL) (*p* = 0.929, 0.506, 0.631, and 0.139, respectively) (Figure 4). Multivariate analysis of adiponectin (<6 μg/mL) and leptin (>4 ng/mL) with initial PSA (>15 ng/mL), clinical Gleason score (≥4), and positive RM showed that adiponectin (<6 μg/mL) and leptin (>4 ng/mL) correlated most strongly with biochemical recurrence among these factors (hazard ratio (HR): 4.03, 95% CI: 1.21–13.5, *p* = 0.021) (Table 5).

Based on the above results, receiver operating characteristic (ROC) curves were generated for patients with prostate cancer and adiponectin of <6 μg/mL and leptin of >4 ng/mL, with the outcome of biochemical recurrence (Figure 5). The sensitivity and specificity of the tests for biochemical recurrence in patients with pT3apN0 disease were 89.2% and 27.3%, respectively.

Next, we broadened the scope of cases to be searched and analyzed the biochemical non-relapse, biochemical relapse (non-CRPC), and CRPC groups of patients. The patient backgrounds of these groups are shown in Table 3. There were 74 patients in the biochemical non-recurrence group, 52 in the biochemical recurrence group (excluding those who progressed to CRPC), and eight in the group who progressed to CRPC. No significant differences were observed in age, surgical technique, BMI, clinical Gleason score (Table 6), medical history, preoperative hormone therapy, observation period, or number of deaths (Table 7) among the three groups; in contrast, significant differences were observed in initial PSA (*p* = 0.001, between the biochemical non-recurrence and biochemical recurrence groups) and clinical T stage (*p* < 0.001, among all groups) (Table 6).

As for pathological factors, significant differences were observed in pathological Gleason score (*p* = 0.001, among all groups), RM (*p* = 0.003, between biochemical non-relapse and biochemical relapse groups and between biochemical non-relapse and CRPC groups), ly (*p* = 0.006, between biochemical non-relapse and CRPC groups and between biochemical relapse and CRPC groups), and v (*p* = 0.020, between biochemical non-relapse and CRPC groups and between biochemical relapse and CRPC groups) (Table 8).

As for serum lipids, no significant differences were observed in preoperative blood TG, T-chol, HDL, and LDL concentrations among the three groups. Similarly, no significant differences were observed in adipokines among the three groups in terms of serum adiponectin and leptin concentrations, adiponectin (<6 μg/mL), and leptin (>4 ng/mL) (Table 9). The A/L ratio tended to decrease from the biochemical non-relapse group to the CRPC group, with no statistically significant difference (*p* = 0.786) (Figure 6). Next, the correlation between androgen receptor expression and the combination of adiponectin and leptin levels was examined by immunostaining of prostate cancer tissue. No significant difference was observed in the androgen receptor positivity in the prostate cancer cells of the adiponectin < 6 µg/mL and leptin > 4 ng/mL group compared to the adiponectin > 6 µg/mL and leptin < 4 ng/mL group (*p* = 0.3696) (Appendix A, Figure 1). As shown in Table 1, there were no significant differences between Gleason scores and disease status-related factors such as T stage or biochemical recurrence. We examined the correlation between adiponectin levels, leptin levels, and Gleason score. The results showed a moderate correlation between adiponectin levels and preoperative Gleason score (*p* = 0.024), but the correlation was very weak (R^2^ = 0.038, Appendix A, Figure 2). No correlation was found between adiponectin levels and postoperative Gleason score or leptin levels and pre- and postoperative Gleason scores (Appendix A, Figure 2). Furthermore, no significant difference was observed in androgen receptor positivity of prostate cancer cells in the adiponectin < 6 and leptin > 4 group compared to the adiponectin > 6 and leptin < 4 group (*p* = 0.3696, Appendix A, Figure 2).

## 3. Discussion

This study analyzed the correlation between serum adipokine concentration and the risks of biochemical recurrence and CRPC progression in fat-invasive prostate cancer. Low levels of preoperative serum adiponectin and leptin were significantly associated with biochemical recurrence.

High-grade prostate cancer with pT3a resulting in EPE is at a high risk of progression to biochemical recurrence [36]; in contrast, a group with a favorable prognosis among EPE cases was reported [37,38,39]. Further studies are needed in patients with pT3a disease and a poor prognosis who will benefit from treatment. Biochemical predictors of pT3a prostate cancer recurrence or progression include the radial distance of extra-prostatic extension [40], and SARIFA was reported to be a pathological biomarker [41]. Because many studies have reported the interaction between cancer and fat contributing to cancer progression [25,26,27,28], we focused on the association between prostate cancer and fat secretion factors and analyzed the correlation between biochemical recurrence and serum adipokine and leptin levels in patients with pT3a pN0 stage disease.

It has been reported that high total cholesterol increases the risk of carcinogenesis of prostate cancer with Gleason score (GS) ≥ 4+3 [42] and high triglyceride [43]. However, when the comparison group was not a control group but a pathological high-grade case, as in the present study, no significant differences were observed in serum lipid concentrations (T-Chol and TG) between the biochemical recurrence and non-recurrence groups (Table 1).

Adiponectin can cause low serum adiponectin and decrease adiponectin receptor expression in the tumor tissues of patients with prostate cancer [44]. In the present study, no significant difference was observed in the serum adiponectin levels between the biochemical recurrence and non-recurrence groups. Similarly, studies have reported the overexpression of leptin receptors in tumor tissues [45,46], but the involvement of leptin itself in recurrence is unknown. In the present study, the serum leptin concentrations were not significantly different between the biochemical relapse and non-relapse groups. Serum concentrations of adiponectin and leptin, which are adipose secretors, correlate with fat mass and may be one of the reasons why these serum concentrations did not correlate with biochemical recurrence alone [15]. In the present study, the blood adiponectin and leptin levels alone did not correlate with biochemical recurrence because the number of patients with a disproportionately high BMI was small. A previous report also examined the correlation of adiponectin and leptin concentrations with prostate cancer progression to CRPC and overall survival (OS) but reported that each alone did not contribute to the correlation, supporting the results of a previous report [47]. 

We then analyzed the correlation between the combinational effect of blood adiponectin and leptin with recurrence, as reported in a study in which the A/L ratio was decreased in women with breast cancer [48] and in a mouse model of TRAMP. Furthermore, the A/L ratio correlated with tumor growth in a mouse model of prostate cancer [49]. Therefore, the A/L ratio may be a useful predictor of the clinical grade of prostate cancer. We hypothesized that a decreased A/L ratio would correlate with cancer recurrence or progression. However, no significant difference was found between the biochemical recurrence and non-recurrence groups. To further investigate the correlation between prostate cancer progression and blood adiponectin and leptin concentrations, we divided the patients into groups with various cutoffs for adiponectin and leptin concentrations and performed statistical analysis to determine whether a correlation occurred with the risk of biochemical recurrence. The results showed that preoperative serum adiponectin (<6 μg/mL) and leptin (>4 ng/mL) significantly increased the risk of biochemical recurrence and was also significant as a risk factor for biochemical recurrence in multivariate analyses of initial PSA, clinical GS, and RM. These observations are consistent with those of previous reports showing that adiponectin and leptin are involved in prostate cancer suppression and growth, respectively [22,26,27,29,50].

Previous meta-analyses have reported that increased adiponectin correlates slightly inversely with prostate cancer prognosis and increased leptin correlates slightly with high-grade prostate cancer; however, most comparisons have been made with Gleason scores or healthy men [51]. Among them, one report [52] correlated adiponectin gene mutations with biochemical recurrence; this is the first report to show a correlation between the adiponectin and leptin combination and biochemical recurrence.

An ROC curve was developed to evaluate the accuracy of biochemical recurrence in patients with adiponectin of <6 μg/mL and leptin of >4 ng/mL; the sensitivity was good, but the specificity was low. Thus, the combination of serum adiponectin and leptin levels in patients with preoperative prostate cancer is a promising predictor of biochemical recurrence.

Because the number of CRPC cases in the pT3a pN0 group was small in this study, we performed an additional analysis to include groups with high-grade pT3b-4 and pN1 CRPC, and the A/L ratio decreased with disease progression. Therefore, it is possible that the A/L ratio may be correlated with the risk of progression to CRPC. Further studies should be conducted with a larger number of patients.

No correlation between biochemical recurrence and BMI was observed in the present analysis or sub-analysis. Previous studies have reported that obesity and BMI are associated with the risk of prostate cancer mortality [53,54]; however, the fact that a clinical correlation was observed between adipokines in this study, regardless of BMI, suggests that this is not simply a relationship with obesity, but rather a correlation with weight, which is not reflected in BMI in this study. The fact that clinical correlations were found between adipokines regardless of BMI in the present study suggests that differences in visceral fat mass relative to body weight and fat quality, such as the ratio of white to brown fat, which is not reflected in BMI, may be involved in the progression of prostate cancer recurrence.

The present study suggests that serum adiponectin and leptin levels are significantly correlated with biochemical recurrence in patients with pT3a pN0 disease. However, EPE is often a small lesion, and serum levels may not necessarily reflect the local environment of the EPE. Future studies are needed to elucidate the mechanism of fat invasion in prostate cancer and identify the targets of drug therapy by observing adiponectin receptor expression in the EPE region.

To examine the clinical significance of adiponectin and leptin levels between the biochemical recurrence and non-biochemical recurrence groups, it was crucial to control for possible confounders. Therefore, the non-biochemical recurrence group was carefully selected to ensure no significant differences in clinical T stage or Gleason score between the two groups. Furthermore, multivariate analysis of adiponectin/leptin levels, PSA, Gleason score, and surgical margins confirmed no significant differences between the groups except for the combination of adiponectin/leptin levels. These findings suggest that potential confounding factors associated with biochemical recurrence, other than adiponectin/leptin levels, were adequately controlled in this study.

This study has several limitations. First, the data were from a single institution, and the number of patients was relatively small. However, it also has an advantage in that the protocols for specimen processing and data management are constant. Most patients at our cancer center present with prostate cancer at a stage suitable for surgery, resulting in a relatively small number of CRPC patients. However, there are nearby hospitals that typically treat a more significant number of CRPC patients, and we plan to conduct a joint study in the future with these hospitals to increase the number of patients and further validate our findings. Second, the number of pT3a pN0 CRPC cases was small because the overall number of cases was small. Therefore, we were unable to perform a valid analysis of the risk of CRPC progression in this study, and we need to increase the number of cases in a multicenter study. Third, serum concentrations may not necessarily reflect adiponectin and leptin expression in the EPE region. In the future, the correlation between serum concentration and EPE will be confirmed in more detail by observing adiponectin and leptin receptor expression at the EPE site. Finally, the intervals between serum collection and surgery were not uniform. In the present study, the time from serum collection to surgery was longer than from preoperative hormone therapy to surgery. Serum was collected within 2–3 months of surgery, but the time between serum collection and surgery was prolonged due to preoperative hormone therapy and surgical waiting time. Future analyses should be conducted with a fixed period between serum collection and surgery.

## 4. Materials and Methods

After obtaining consent for the Institutional Review Board (IRB)-approved study plan, we included 99 patients in the patient group who underwent pancreatectomy for pT3a pN0 stage prostate cancer at our institution between 2011 and 2019 to determine the clinical correlation between blood adipokine levels and biochemical recurrence or castration resistance. Patients with pT3b, pT4, and pN1 were excluded from this study because the stage of the disease itself seemed to contribute more to the risk of recurrence.

The primary outcome was biochemical recurrence (PSA ≥ 0.2). However, since only a small number of the 99 patients with pT3apN0 progressed to CRPC, we conducted a sub-analysis of 134 patients, including 35 patients with pT3b, pT4, and pN1, who were considered exclusion criteria, to determine blood adipokine levels and the risk of CRPC in three groups: the biochemical non-relapse group, the biochemical relapse group (non-CRPC), and the CRPC group. Correlations among blood adipokine levels, biochemical recurrence, and CRPC progression were analyzed.

### 4.1. Clinical Information 

Clinical information used for analysis included age, body mass index (BMI), surgical procedure (retropubic radical prostatectomy (RRP), laparoscopic radical prostatectomy (LRP), robot-assisted laparoscopic radical prostatectomy (RARP), initial PSA, preoperative TG, T-chol, HDL, LDL, blood adipokine levels (adiponectin, leptin, and adiponectin/leptin ratio) clinical T stage, clinical Gleason score, HT clinical Gleason score, HT, HDL, history of DM, preoperative hormone therapy, pathological factors (pathological Gleason score, RM, ly, v, and pn ) of prostate cancer tissues that were totally resected, observation periods, and outcomes.

### 4.2. Detection of Serum Adipokine Concentration

After obtaining comprehensive informed consent from patients, we measured adiponectin and leptin levels in sera from 99 eligible patients; the serum samples were stored at the Kanagawa Cancer Center Biospecimen Center. Preoperative sera were collected during a blood test immediately before surgery. All serum samples were sent to the Special Reference Laboratory (SRL) International Inc. (Tokyo, Japan) for adipokine detection. At the SRL, adiponectin concentrations were measured using a chemiluminescence enzyme immunoassay, and leptin concentrations were measured using a double-antibody radioimmunoassay.

### 4.3. Statistical Analysis

Fisher's exact probability test and the Chi-square test were used to compare the means of continuous variables, and Fisher's exact probability test and the Kruskal–Wallis test were used to compare the proportions of categorical variables. Differences were considered statistically significant at *p* < 0.05. Conditional logistic regression models were used to estimate odds ratios (ORs) and 95% confidence intervals (CIs) to evaluate the association of each variable with the biochemical recurrence of prostate cancer. All statistical analyses were performed using EZR (Saitama Medical Center, Jichi Medical University, Saitama, Japan), a graphical user interface for R (The R Foundation for Statistical Computing, Vienna, Austria), and GraphPad Prism version 10.1.2(324) [55].

### 4.4. Ethics

This study was approved by the Institutional Review Board of the Kanagawa Cancer Center (IRB number: 2020EKI37). It conformed to the provisions of the Declaration of Helsinki and Ethical Guidelines for Life Sciences and Medical Research Involving Human Subjects of the Japanese Personal Data Protection Act.

## 5. Conclusions

This study analyzed the correlation between preoperative serum adipokine levels and the risks of biochemical recurrence and CRPC progression in patients with fat-invasive prostate cancer. We found that the combination of serum adiponectin and leptin levels was significantly associated with biochemical recurrence. Future studies will focus on analyzing adiponectin and leptin receptor expression in the region of fatty infiltration, leading to the screening of patients with a poor prognosis and adipokine receptor-targeted therapy. These findings could help develop new approaches to improve prostate cancer diagnosis and treatment. Our study is a pilot study, and additional retrospective and prospective studies with increased cases are needed to validate the present results further.

## Figures and Tables

**Figure 1 ijms-25-08970-f001:**
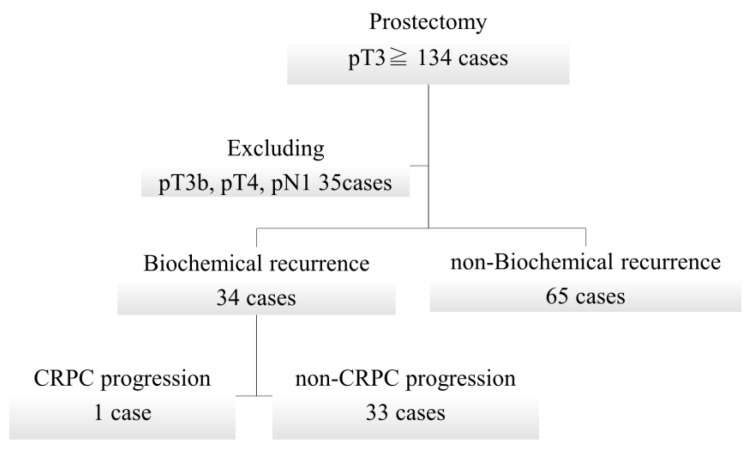
Patient selection flow chart. A total of 134 patients underwent total prostatectomy and had pathological phenotypes of pT3 or higher stages of the disease. Of these patients, 35 had pT3b, pT4, or pN0 stages of the disease and were excluded, 34 had biochemical recurrences, and 65 had non-recurrences. Of the 34 biochemical recurrences, one developed CRPC. CRPC, castration-resistant prostate cancer.

**Figure 2 ijms-25-08970-f002:**
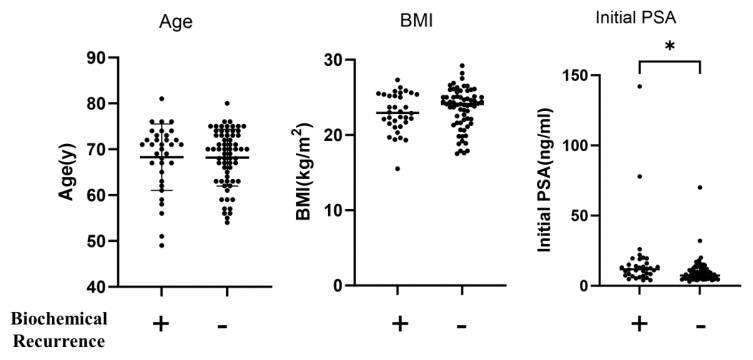
Analysis of differences in age, body mass index, and initial PSA between patients with and without biochemical recurrence. Age (left), BMI (middle), and initial PSA (right) were compared between patients with (*n* = 34) and without (*n* = 65) biochemical recurrence. Statistical analyses were performed using unpaired t-tests. BMI, body mass index; y, year. * *p* < 0.05.

**Figure 3 ijms-25-08970-f003:**
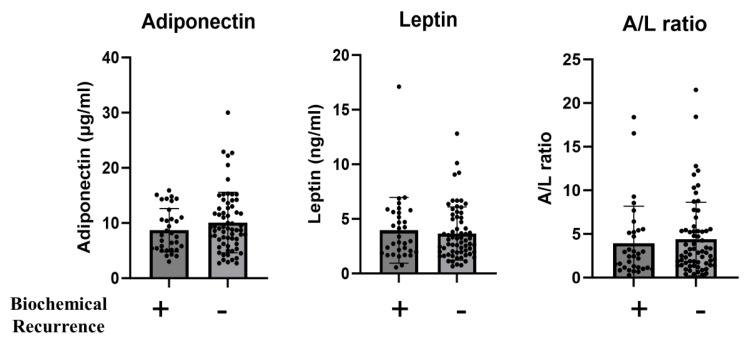
Analysis of differences in preoperative serum adiponectin and leptin concentrations and adiponectin/leptin ratios between patients with and without biochemical recurrence. Serum adiponectin concentration (left), serum leptin concentration (middle), and A/L ratio (right) were compared between patients with (*n* = 34) and without biochemical recurrence (*n* = 65). Statistical analyses were performed using the unpaired t-test. A/L ratio, adiponectin/leptin ratio.

**Figure 4 ijms-25-08970-f004:**
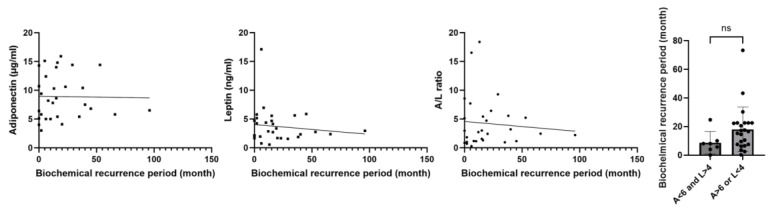
Analysis of differences in preoperative serum adiponectin (left), leptin (middle), and adiponectin/leptin ratios (right) in patients with biochemical recurrence and the duration for biochemical recurrence. We compared the correlation between the duration from surgery to biochemical recurrence and preoperative serum adiponectin concentration, leptin concentration, A/L ratio, or the presence of adiponectin (<6 μg/mL) and leptin (>4 ng/mL) among patients (*n* = 34) who developed biochemical recurrence. Statistical analyses were performed using the unpaired t-test. A/L ratio, adiponectin/leptin ratio. ns: not significant (*p* > 0.05).

**Figure 5 ijms-25-08970-f005:**
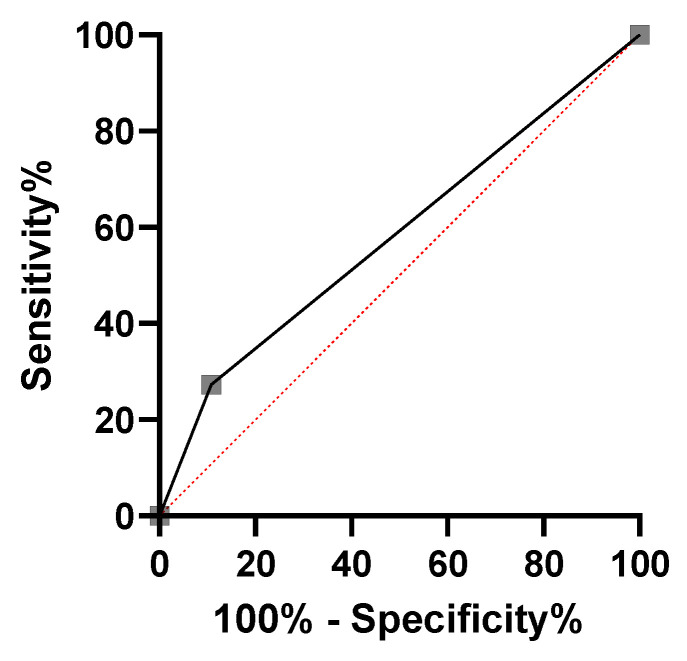
Receiver operating characteristic curves for a population of 99 patients with a pT3a pN0 stage of the disease with preoperative adiponectin of <6 μg/mL and leptin of >4 ng/mL. The sensitivity and specificity for biochemical recurrence in the population of patients were 89.2% and 27.3%, respectively. ROC, receiver operating characteristic. Black solid line: ROC for predicting biochemical recurrence; red dotted line: reference line.

**Figure 6 ijms-25-08970-f006:**
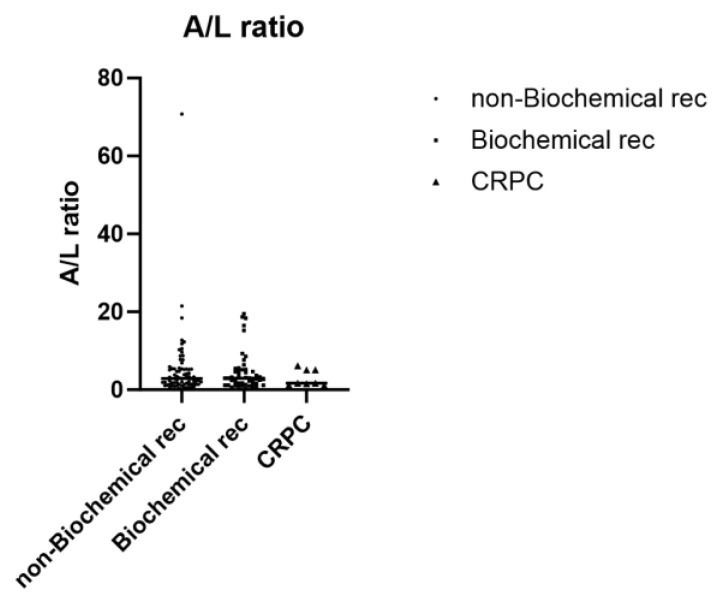
Analysis of differences in preoperative adiponectin/leptin ratios among cases without biochemical recurrence, cases with biochemical recurrence and without CRPC, and cases with CRPC. Preoperative A/L ratios were compared among the three groups of patients without biochemical recurrence (*n* = 74), with biochemical recurrence and without CRPC (*n* = 52), and cases with CRPC (*n* = 8). Statistical analyses were performed using the Kruskal–Wallis test. Rec, recurrence; A/L ratio, adiponectin/leptin ratio; and CRPC, castration-resistant prostate cancer.

**Table 1 ijms-25-08970-t001:** Characteristics of all patients.

	Non-Biochemical Rec (*n* = 65)	Biochemical Rec (*n* = 34)	*p* Value
Age, median (min, max) y	70 (54–80)	71 (49–81)	0.748
Operation			0.654
RRP	3 (4.6)	1 (2.9)	
LRP	24 (58.5)	16 (47.1)	
RARP	38 (36.9)	17 (50)	
BMI, median (min, max) kg/m^2^	24.1 (17.5–29.2)	22.95 (15.5–27.3)	0.323
Initial PSA (ng/mL)	7.4 (3.1–70)	11.85 (4.1–142)	0.006
Clinical T stage, No (%)			0.184
T1	6 (9.2)	2 (5.9)	
T2	50 (77)	20 (58.8)	
T3	9 (13.8)	12 (35.3)	
Clinical Gleason score, No (%)			0.235
6	2 (3.1)	2 (5.9)	
7	32 (49.2)	10 (29.4)	
8	23 (35.4)	16 (47.1)	
9	8 (12.3)	6 (17.6)	

Rec, recurrence; RRP, retropubic radical prostatectomy; LRP, laparoscopic radical prostatectomy; and RARP, robotic-assisted radical prostatectomy.

**Table 2 ijms-25-08970-t002:** History, duration of treatment, and prognosis of patients.

	Non-Biochemical Rec (*n* = 65)	Biochemical Rec (*n* = 34)	*p* Value
Hypertension, No (%)	24 (36.9)	12 (35.3)	1
Dyslipidemia, No (%)	16 (24.6)	9 (26.5)	1
Diabetes, No (%)	7 (10.8)	5 (14.7)	0.747
Neoadjuvant hormone therapy, No (%)	7 (10.8)	6 (17.6)	0.36
Follow-up period, month (min, max)	23 (0–98)	39 (3–145)	<0.001
Dead, No (%)	1 (1.5)	2 (5.9)	0.271

Rec, recurrence.

**Table 3 ijms-25-08970-t003:** Pathological outcomes of patients.

	Non-Biochemical Rec (*n* = 65)	Biochemical Rec (*n* = 34)	*p* Value
Pathological Gleason score, No (%)			0.29
≦7	52 (79.7)	21 (61.7)	
8	9 (13.8)	9 (26.5)	
9	4 (6.2)	4 (11.8)	
RM, No (%)	28 (43.8)	21 (63.6)	0.087
ly, No (%)	9 (13.8)	2 (5.9)	0.322
v, No (%)	7 (10.8)	1 (2.9)	0.257
pn, No (%)	62 (95.4)	31 (91.2)	0.41
Salvage radiation therapy, No (%)	0 (0)	26 (76.5)	<0.001
CRPC progression, No (%)	0 (0)	1 (2.9)	0.343

RM, radial margin; Rec, recurrence; ly, lymphatic vessel invasion; v, venous invasion; pn, perineural invasion; and CRPC, castration-resistant prostate cancer.

**Table 4 ijms-25-08970-t004:** Blood adipokine and lipid levels in patients.

	Non-Biochemical Rec (*n* = 65)	Biochemical Rec (*n* = 34)	*p* Value
Adiponectin, median (min, max), μg/mL	9 (2.7–30)	7.8 (3–15.9)	0.328
Leptin, median (min, max), ng/mL	3.02 (0.66–12.80)	2.95 (0.56–17.10)	0.647
Adiponectin/Leptin(A/L) ratio, median (min, max)	2.92 (0.33–21.52)	2.45 (0.29–18.39)	0.323
Adiponectin < 6 and Leptin > 4, No (%)	7 (10.8)	9 (27.3)	0.046
Adiponectin < 7 and Leptin > 4, No (%)	8 (12.3)	10 (29.4)	0.051
Triglyceride, median (min, max) mg/dL	111.5 (34–1509)	139 (51–383)	0.316
Total Cholesterol, median (min, max) mg/dL	211 (155–318)	211 (156–272)	0.993
HDL, median (min, max) mg/dL	59 (42–111)	57 (42–90)	0.868
LDL, median (min, max) mg/dL	114 (62–222)	115.8 (56–186)	0.527

Rec, recurrence; HDL, High-Density Lipoprotein; LDL, Low-Density Lipoprotein; and No, number.

**Table 5 ijms-25-08970-t005:** Multivariate analysis of adipokine levels, initial PSA, and pathological findings.

	Hazard Ratio	Lower Confidence Interval	Upper Confidence Interval	*p* Value
Adiponectin < 6 and Leptin > 4	4.03	1.23	13.2	0.021
Initial PSA (ng/mL) > 15	2.11	0.69	6.46	0.192
Clinical Gleason ≧ 4	1.93	0.75	4.99	0.176
Surgical margin positive	1.82	0.73	4.59	0.201

PSA, prostate-specific antigen.

**Table 6 ijms-25-08970-t006:** Characteristics of patients recategorized including CRPC progression.

	Non-Biochemical Rec (*n* = 74)	Biochemical Rec (*n* = 52)	CRPC Progression (*n* = 8)	*p* Value
Age, median (min, max) y	69 (54–80)	70 (49–81)	67 (66–72)	0.943
Operation				0.925
RRP	4 (5.4)	2 (3.8)	0 (0)	
LRP	28 (37.8)	20 (38.5)	4 (50)	
RARP	42 (56.8)	30 (57.7)	4 (50)	
BMI, median (min, max) kg/m^2^	24 (17.5–29.2)	23.05 (15.5–30.9)	23.4 (17.2–29)	0.538
Initial PSA (ng/mL)	7.6 (3.1–70)	11.85 (4.1–142)	9 (6.3–21)	0.001
Clinical T stage, No (%)				<0.001
T ≦ 2	65 (86.5)	30 (57.7)	2 (25)	
T3	9 (13.5)	22 (42.3)	4 (50)	
T4	0 (0)	0 (0)	2 (25)	
Clinical Gleason score, No (%)				0.215
≦7	38 (51.4)	18 (34.6)	2 (25)	
8	27 (36.4)	19 (36.5)	3 (37.5)	
≧9	9 (12.2)	15 (28.9)	3 (37.5)	

Rec, recurrence; RRP, retropubic radical prostatectomy; LRP, laparoscopic radical prostatectomy; RARP, robotic-assisted radical prostatectomy; BMI, body mass index; PSA, prostate-specific antigen; and No, number.

**Table 7 ijms-25-08970-t007:** History, duration of treatment, and prognosis of patients recategorized including CRPC progression.

	Non-Biochemical Rec (*n* = 74)	Biochemical Rec (*n* = 52)	CRPC Progression (*n* = 8)	*p* Value
Hypertension, No (%)	26 (35.1)	17 (32.7)	2 (25)	0.834
Dyslipidemia, No (%)	17 (23)	12 (23.1)	2 (25)	0.992
Diabetes, No (%)	8 (10.8)	7 (13.5)	3 (37.5)	0.11
Neoadjuvant hormone therapy, No (%)	10 (13.5)	10 (19.2)	2 (25)	0.553
Follow-up period, month (min, max)	23.5 (0–112)	36 (3–145)	55 (19–89)	0.128
Dead, No (%)	1 (1.4)	2 (3.8)	1 (12.5)	0.191

Rec, recurrence; CRPC, castration-resistant prostate cancer; and No, number.

**Table 8 ijms-25-08970-t008:** Pathological outcomes recategorized including CRPC progression.

	Non-Biochemical Rec (*n* = 74)	Biochemical Rec (*n* = 52)	CRPC progression (*n* = 8)	*p* Value
Pathological Gleason score, No (%)				0.001
≦7	59 (79.7)	26 (50)	3 (37.5)	
8	9 (12.2)	14 (26.9)	0 (0)	
9	6 (8.1)	12 (23.1)	5 (62.5)	
RM, No (%)	30 (41.1)	36 (70.6)	6 (75)	0.003
ly, No (%)	11 (14.9)	11 (21.2)	5 (62.5)	0.006
v, No (%)	9 (12.2)	8 (15.4)	4 (50)	0.02
pn, No (%)	71 (95.9)	49 (94.2)	8 (100)	0.738
Salvage radiation therapy, No (%)	0 (0)	39 (75)	3 (37.5)	<0.001

Rec, recurrence; CRPC, castration-resistant prostate cancer.; RM, radial margin; ly, lymphatic vessel invasion; v, venous invasion; and pn, perineural invasion.

**Table 9 ijms-25-08970-t009:** Blood adipokine and lipid levels in patients recategorized including CRPC progression.

	Non-Biochemical Rec (*n* = 74)	Biochemical Rec (*n* = 52)	CRPC Progression (*n* = 8)	*p* Value
Adiponectin, median (min, max) μg/mL	8.75 (2.7–30)	7.65 (3–40)	7.8 (5.9–21.5)	0.672
Leptin, median (min, max) ng/mL	3.15 (0.27–12.8)	2.81 (0.56–17.1)	4.28 (1.04–12.8)	0.46
Adiponectin/Leptin(A/L) ratio, median (min, max)	2.9 (0.33–70.7)	2.96 (0.29–19.51)	1.69 (1.16–6.15)	0.786
Adiponectin < 6 and Leptin > 4, No (%)	9 (12.2)	10 (20)	0 (0)	0.232
Triglyceride, median (min, max) mg/dL	109.5 (34–1509)	135 (51–383)	91 (80–158)	0.128
Total cholesterol, median (min, max) mg/dL	211 (155–318)	208.5 (156–272)	214 (158–251)	0.981
HDL, median (min, max) mg/dL	60 (42–111)	58.5 (29–97)	62.5 (29–92)	0.831
LDL, median (min, max) mg/dL	114 (62.4–222)	107.4 (56–187.2)	114 (84.6–161.4)	0.606

Rec, recurrence; CRPC, castration-resistant prostate cancer; HDL, High-Density Lipoprotein; LDL, Low-Density Lipoprotein; and No, number.

## Data Availability

The data presented in this study are available upon request from the corresponding author.

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
