# Peer review of "Combination of Blood Adiponectin and Leptin Levels Is a Predictor of Biochemical Recurrence in Prostate Cancer Invading the Surrounding Adipose Tissue"

_ijms, 2024, doi:10.3390/ijms25168970_

Round 1
Reviewer 1 Report
Comments and Suggestions for Authors
Dear authors,
ijms-3134396
Combination of Blood Adiponectin and Leptin Levels is a Predictor of Biochemical Recurrence in Prostate Cancer Invading 3 the Surrounding Adipose Tissue by Suzuki et al describes retrospectively analyzed the clinical data, including preoperative serum adipokine levels, of 99 patients with pT3a pN0 19 prostate cancer who underwent proctectomy between 2011 and 2019. The primary outcome was 2biochemical recurrence (prostate specific antigen: PSA>0.2). We identified 65 non-recurrences and 34 biochemical recurrences (one progressed to CRPC). The initial PSA level was significantly 22 higher (p=0.006), but serum adiponectin (p=0.328) and leptin (p=0.647) levels and their ratio 23 (p=0.323) were not significantly different in the biochemical recurrence group compared with in the non-recurrence group. In contrast, significantly more biochemical recurrences were observed in the group with adiponectin<6 μg/ml and Leptin<4 ng/ml (p=0.046), initial PSA>15 ng/ml. This is interesting study. However, this needs additional data to strengthen the manuscript.
Comments:
1. What is the AR status in adiponectin<6 μg/ml and Leptin<4 ng/ml patients.
2. Does the diseases status and adiponectin and Leptin levels correlate with Gleason score.
3. The study should involve more patients.
Author Response
Comment 1:
What is the AR status in adiponectin<6 μg/ml and Leptin<4 ng/ml patients.
Response 1:
Thank you for your important question. We compared the expression of androgen receptors in prostate cancer tissue in all 16 cases where the adiponectin/leptin combination levels correlated with biochemical recurrence and 25 cases with the opposite level combinations. The results showed no significant differences. We have now added this information to the revised manuscript.
Results, page 11, lines 238 to 243:
"Next, the correlation between androgen receptor expression and the combination of adiponectin and leptin levels was examined by immunostaining of prostate cancer tissue. No significant difference was observed in the androgen receptor positivity in the prostate cancer cells of the adiponectin < 6 µg/mL and leptin > 4 ng/mL group compared to the adiponectin > 6 µg/mL and leptin < 4 ng/mL group (p=0.3696) (Supplementary Figure 1)."
Comment 2:
Does the diseases status and adiponectin and Leptin levels correlate with Gleason score.
Response 2:
Thank you for the valuable question. We have added information on these parameters to the revised manuscript.
Results, page 11, lines 243 to 252:
“As shown in Table 1, there were no significant differences between Gleason scores and disease status-related factors such as T stage or biochemical recurrence. We examined the correlation between adiponectin levels, leptin levels, and Gleason score. The results showed a moderate correlation between adiponectin levels and preoperative Gleason score (p=0.024), but the correlation was very weak (R2 = 0.038, Supplementary Figure 2). No correlation was found between adiponectin levels and postoperative Gleason score or leptin levels and pre- and postoperative Gleason scores (Supplementary Figure 2). Furthermore, no significant difference was observed in androgen receptor positivity of prostate cancer cells in the adiponectin <6 and leptin >4 group compared to the adiponectin >6 and leptin <4 group (p=0.3696, Supplementary Figure 2).”
Comment 3:
The study should involve more patients.
Response 3:
Thank you for the valuable suggestion. We acknowledge the limitations related to the relatively small number of CRPC patients in our study. Unfortunately, we cannot increase the number of patients at this stage as our cancer center only receives few such patients. However, to address this limitation, we have outlined our plans for a collaborative study with nearby hospitals that treat a larger population of CRPC patients. This future collaboration will allow us to increase the sample size and further validate our findings, thereby strengthening the study’s conclusions. We have now added this information to the revised manuscript.
Discussion, page 13, lines 359-364:
“Most patients at our cancer center present with prostate cancer at a stage suitable for surgery, resulting in a relatively small number of CRPC patients. However, there are nearby hospitals that typically treat a more significant number of CRPC patients, and we plan to conduct a joint study in the future with these hospitals to increase the number of patients and further validate our findings.”

Reviewer 2 Report
Comments and Suggestions for Authors
The manuscript addressed the correlation between blood adipokine levels, specifically adiponectin and leptin, and the risk of biochemical recurrence in patients with prostate cancer that has invaded surrounding adipose tissue. The study analyzed clinical data from 99 patients and identified a significant association between low preoperative serum adiponectin and high leptin levels with biochemical recurrence. The main study design is clear and well presented. The description is generally good. My overall comment would be accept with minor revision. The detailed comments are as follows:
1. Can you elaborate on the novelty of your findings in relation to previous related works in the introduction section? Specifically, how do your results differ from or expand upon current understandings of adiponectin and leptin's roles in prostate cancer? And what is the
- You mentioned the retrospective analysis. Can you provide more details on controling confounding factors?
3. Other two questions are listed as limitations in the discussion part with future plan. Larger sample size would be one of my suggestions to the future work.
Author Response
Reviewer 2
Comment 1:
Can you elaborate on the novelty of your findings in relation to previous related works in the introduction section? Specifically, how do your results differ from or expand upon current understandings of adiponectin and leptin's roles in prostate cancer?
Response 1:
Thank you for your question. We have expanded the Introduction to highlight the novelty of our study. While previous research has explored the roles of adiponectin and leptin in various aspects of prostate cancer, our study is the first to combine these adipokines as predictive markers of biochemical recurrence. This approach aims to enhance the accuracy of predicting this important clinical outcome.
Introduction, pages 2 to 3, lines 68 to 73:
“The combined clinical significance of adiponectin and leptin in prostate cancer patients has been studied in various contexts, including risk of prostate carcinogenesis (PMID 22971685), patient obesity and abnormal glucose metabolism (PMIDs 38299570, 37794972, and 33295204), adipokine expression in prostate tissue (PMID38537999), and correlation with cardiovascular disease (PMID37917926). However, this is the first study to combine adiponectin and leptin as predictive markers of biochemical recurrence, aiming for a more accurate prediction of this outcome.”
Comment 2:
You mentioned the retrospective analysis. Can you provide more details on controling confounding factors?
Response 2:
Thank you for the suggestion. We have revised the manuscript to provide a detailed explanation of how we controlled for potential confounding factors in our analysis. Specifically, we ensured that the non-biochemical recurrence group was matched to the biochemical recurrence group in terms of clinical T stage and Gleason score. Additionally, multivariate analysis confirmed that no significant differences were present except for the combination of adiponectin/leptin levels. We believe these steps effectively controlled for confounding factors, and the revised text has been added to the manuscript.
Discussion, page 13, lines 348 to 356:
“To examine the clinical significance of adiponectin and leptin levels between the biochemical recurrence and non-biochemical recurrence groups, it was crucial to control for possible confounders. Therefore, the non-biochemical recurrence group was carefully selected to ensure no significant differences in clinical T stage or Gleason score between the two groups. Furthermore, multivariate analysis of adiponectin/leptin levels, PSA, Gleason, and surgical margins confirmed no significant differences between the groups except for the combination of adiponectin/leptin levels. These findings suggest that potential confounding factors associated with biochemical recurrence, other than adiponectin/leptin levels, were adequately controlled in this study.”
Comment 3:
Other two questions are listed as limitations in the discussion part with future plan. Larger sample size would be one of my suggestions to the future work.
Response 3:
Thank you very much for the crucial suggestion. We acknowledge the limitations related to the relatively small number of CRPC patients in our study. However, to address this limitation, we have outlined our plans for a collaborative study with nearby hospitals that treat a larger population of CRPC patients. This future collaboration will allow us to increase the sample size and further validate our findings, thereby strengthening the study’s conclusions. We have now added this information to the revised manuscript.
Discussion, page 13, lines 359-364:
“Most patients at our cancer center present with prostate cancer at a stage suitable for surgery, resulting in a relatively small number of CRPC patients. However, there are nearby hospitals that typically treat a more significant number of CRPC patients, and we plan to conduct a joint study in the future with these hospitals to increase the number of patients and further validate our findings.”

Round 2
Reviewer 1 Report
Comments and Suggestions for Authors
The manuscript is improved
Author Response
Point-by-point response to the reviewer 1
Reviewers' Comments to Author:
Reviewer 1
Comment 1: The manuscript is improved
Response 1: Thank you very much for the meaningful suggestions in the previous review.
We sincerely appreciate your helpful suggestions that have improved the manuscript.